# Reawakening knowledge: Anticipatory recovery from catastrophic interference via structured training

**Yanlai Yang**[1]**, Matt Jones**[2]**, Michael C. Mozer**[3,2]**, and Mengye Ren**[1]

[1]New York University, [2]University of Colorado, Boulder, [3]Google DeepMind
{yy2694,mengye}@nyu.edu, mcj@colorado.edu, mcmozer@google.com

## Abstract

We explore the training dynamics of neural networks in a structured non-IID setting where documents are presented cyclically in a fixed, repeated sequence. Typically, networks suffer from catastrophic interference when training on a sequence of documents; however, we discover a curious and remarkable property of LLMs finetuned sequentially in this setting: they exhibit *anticipatory* behavior, recovering from the forgetting on documents *before* encountering them again. This behavior occurs even though the documents are never presented in context together. The behavior emerges and becomes more robust as the architecture scales up its number of parameters. Through comprehensive experiments and visualizations, we demonstrate a new mechanism by which over-parametrized neural networks can recover from catastrophic interference and uncover new insights into training over-parameterized networks in cyclically structured environments.

## 1 Introduction

Large language models (LLMs) [1, 2, 3, 4] have demonstrated remarkable general capabilities in a wide range of natural language tasks. During the training of LLMs, documents are typically uniformly sampled at random. Due to the large scale of the training set—in contrast to many other domains—LLM training typically occurs in an online fashion: each document is used only once for just one update step without further repetition [5, 6, 7].

Such a training style is in stark contrast with how real world agents like humans acquire new knowledge. In naturalistic settings, the material we are exposed to is structured in time and often repeats in predictable, quasi-cyclic patterns (e.g., a person's everyday morning routine consists of first taking a shower, then eating breakfast, and finally dressing up). Hence it is important to understand how existing deep learning methods and architectures perform in this setting.

Toward the goal of investigating more naturalistic training setups, we study a simplistic setting involving structured training of LLMs: documents are presented cyclically in a fixed sequence and repeated multiple times, just as we humans go through our daily routines. Moreover, to account for the cost of switching among documents (analogous to the mental switching cost between different environments and the waiting cost of obtaining new data), we allow the network to take multiple gradient steps for each document. Compared to standard task-incremental and class-incremental continual learning settings [8] which experience each task only once, our cyclic training setting better approximates the quasi-cyclic temporal structure of real-world environments.

Typically, networks exhibit *catastrophic interference* (also known as catastrophic forgetting) [9] when training on a sequence of tasks: the loss on a given document increases as the training advances to other documents. Curiously, we discover that in a structured training environment, LLMs exhibit a remarkable *anticipatory recovery* behavior: they recover from the forgetting of

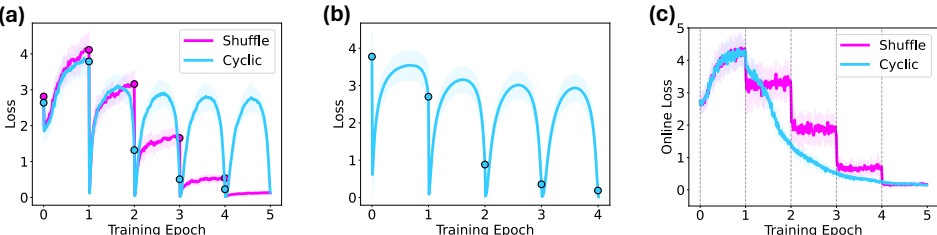

Figure 1: (a) Loss curves on document 1 for cyclic and random shuffled fine-tuning on a pre-trained Pythia-1B model. The black circles indicate points just prior to training on the focal document. The inverted-U loss curves within each epoch demonstrate the anticipatory recovery phenomenon. (b) Shift-averaged loss curve for cyclic fine-tuning. (c) Online loss curves for cyclic and random shuffled fine-tuning with prequential evaluation.

| Model Size | 410M | 1B | 1.4B | 2.8B |
|---|---|---|---|---|
| Cyclic | **1.09 ± 0.03** | **1.03 ± 0.03** | **1.14 ± 0.06** | **1.34 ± 0.06** |
| Shuffle | 1.34 ± 0.02 | 1.51 ± 0.04 | 1.51 ± 0.04 | 1.79 ± 0.06 |

Table 1: Average online loss across epochs 2 to 5 for cyclic fine-tuning and random shuffled fine-tuning.

one document before seeing it again, multiple steps in the sequence prior to the recurrence of the document (see Figure 1(a) and 1(b)). It is analogous to a person anticipating to eat breakfast while taking a morning shower, but leaving the thought aside for the rest of the day. Critically, we never present two documents together in context, so the model cannot directly learn sequential relationships between them. Thus, our finding is surprising as there is no explicit memory in LLMs that stores sequential knowledge across context windows, and there is no systematic overlap of content across documents—the behavior emerges from a random document sequence after repeated exposure to that sequence. Furthermore, we demonstrate that, as a result of anticipatory recovery, training with fixed ordering achieves superior performance than random shuffling in the prequential evaluation [10] setting (see Figure 1(c) and Table 1). For an agent that continuously acts and learns in the real world, the performance on the upcoming task is what matters, and prequential evaluation measures such performance. This result hints at the practical benefits of structured training.

Through extensive experiments, we study how different factors in model architecture and training contribute to the anticipatory recovery phenomenon (Section 3.3). We show that only large-scale networks exhibit this reawakening of knowledge, and smaller ones exhibit no such behavior (Section 3.2). We also show that this phenomenon is not unique to LLMs; some vision models with sufficient width and depth also demonstrate a similar behavior, but LLMs on language modeling tasks exhibit the strongest recovery (Section 3.4). We offer insights on the training dynamics in sequentially and cyclically structured input data and propose hypotheses for the causes of the behavior (Section 4).

## 2 Data and Experiment Setup

In this section, we describe the models, datasets, and training setups that we use in the subsequent experiments. Additional details are presented in Appendix A.

**Models.** For the LLM experiments, we use Pythia [11], a suite of decoder-only autoregressive language models pre-trained on the Pile dataset [12, 13]. We use pre-trained Pythia models ranging from 160M to 2.8B parameters. For the vision experiments, we use pre-trained Image GPT [14] models for causal image modeling and pre-trained vision transformer (ViT) [15] and VGG-19 [16] models for image classification.

**Datasets.** For the LLM experiments, we use the CNN/Daily Mail news dataset [17]. We repurpose it for causal language modeling by discarding the summaries and only using the articles. Importantly, the CNN/Daily Mail dataset is not part of the Pile dataset and hence it is a new domain for the Pythia pre-trained models. We use the same documents for both training and evaluation. Our goal here is not to determine whether a trained model generalizes to new documents, but rather to study the memory for a particular document as a function of position within the training history. For the vision experiments, we use images sampled from CIFAR-10 [18] and ImageNet [19].

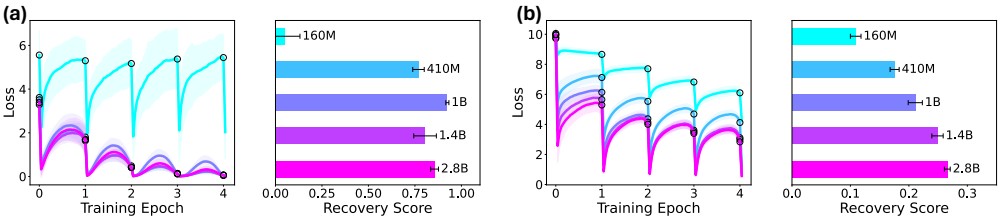

Figure 2: Effect of model size for (a) pre-trained models and (b) random initializations. In each subfigure, the left shows shift-averaged loss curves and the right shows the recovery score as a function of model size.

**Training Setup.** We randomly sample $T$ documents from the dataset. In pre-processing, we truncate each document to the first $C$ tokens (we refer to $C$ as "context length" in subsequent text). We then fine-tune the LLM on each pre-processed sample for $M$ gradient steps (i.e., using a batch size of 1). We refer to the multiple gradient updates of each document as an "episode". After each episode we evaluate the loss on all $T$ documents. We repeat the training process for $E$ epochs, where an epoch consists of one episode of each document in a fixed sequence. We use a vanilla gradient descent optimizer. Unless otherwise stated, the default hyperparameters in the subsequent experiments are $T = 25$, $C = 256$, $M = 10$, $E = 5$. We use the average cross entropy loss (average negative log-likelihood for each token) as our training and evaluation metric.

## 3 Emergent Anticipatory Recovery

In this section, we present our experimental results that reveal the anticipatory recovery phenomenon in cyclic fine-tuning of large language models. We then demonstrate that anticipatory recovery is an emergent behavior that appears only for models with sufficient capacity.

### 3.1 The Anticipatory Recovery Phenomenon

In this first experiment, we have $T = 100$ documents, and we do cyclic fine-tuning of a pre-trained Pythia-1B model [11] on the documents for $E = 5$ epochs in the same ordering. Both the documents and the ordering are sampled at random beforehand, but kept fixed during the sequential fine-tuning process. We refer to these $T$ documents as $x_1, \cdots, x_T$. At the start, we fine-tune on $x_1$ for $M = 10$ gradient steps, leading to a significant decrease in the model's loss on $x_1$. As we move away from $x_1$ and fine-tune on other documents, we naturally observe catastrophic interference: the model's loss on $x_1$ gradually increases until we finish fine-tuning on all other documents and return to $x_1$. As we iterate through the same document sequence for a second time, we would normally expect the loss on $x_1$ to increase monotonically after the initial decrease. However, Figure 1(a) shows that the loss on $x_1$ peaks around $x_{60}$ and then starts to decrease. Before we return to $x_1$, the model has recovered more than half of its initial forgetting during the second epoch. We refer to this counterintuitive decrease in loss as the *anticipatory recovery* phenomenon. In Figure 1(b), we plot the losses for all the documents and re-align them so that 0 on the x-axis refers to the loss on each document $t$ immediately before training on it for the first time. The figure confirms that the anticipatory recovery phenomenon exists for not only $x_1$ but all documents. On the other hand, when we randomly shuffle the document order within each epoch (except $x_1$ is always the first document), we do not observe such anticipatory recovery behavior, and the loss on $x_1$ keeps increasing before we return to it every time.

To quantify the strength of the anticipatory recovery phenomenon, we define the *recovery score* as the proportion of the initial forgetting during the current epoch that the model recovers before returning to the same document. Mathematically, let the mean (over $t$) of the maximum loss on each document $x_t$ between the $n^{\text{th}}$ and $(n+1)^{\text{th}}$ time we train on that document be $l_{\max}(n)$, right before the $(n+1)^{\text{th}}$ time we train on it be $l_{\text{before}}(n)$, and right after the $(n+1)^{\text{th}}$ time we train on it be $l_{\text{after}}(n)$. Then we define the recovery score (RS) for epoch $n$ to be[1] $RS(n) = \frac{l_{\max}(n) - l_{\text{before}}(n)}{l_{\max}(n) - l_{\text{after}}(n-1)}$. In the following

---

[1] In some cases, a randomly initialized model will produce loss curves that decrease throughout the epoch, because its knowledge is so poor that it enjoys positive generalization among all documents. This yields a misleadingly large recovery score under this definition. We do not include such cases in our experiments so do not bother with more nuanced recovery scores.

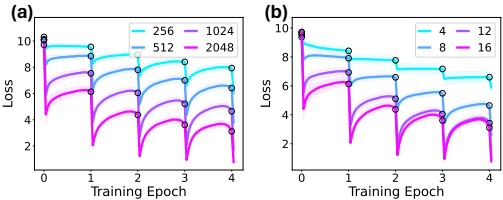

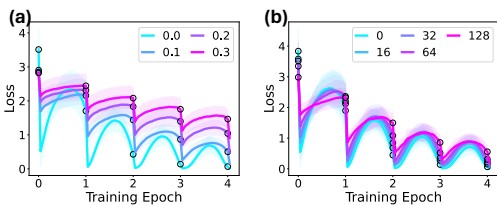

Figure 3: Models trained from scratch with (a) different width (token embedding size) and (b) different depth (number of transformer blocks).

Figure 4: Effect of data randomization strength. (a) Random masking with probability up to 0.3; (b) Random shift of context window up to 128 tokens.

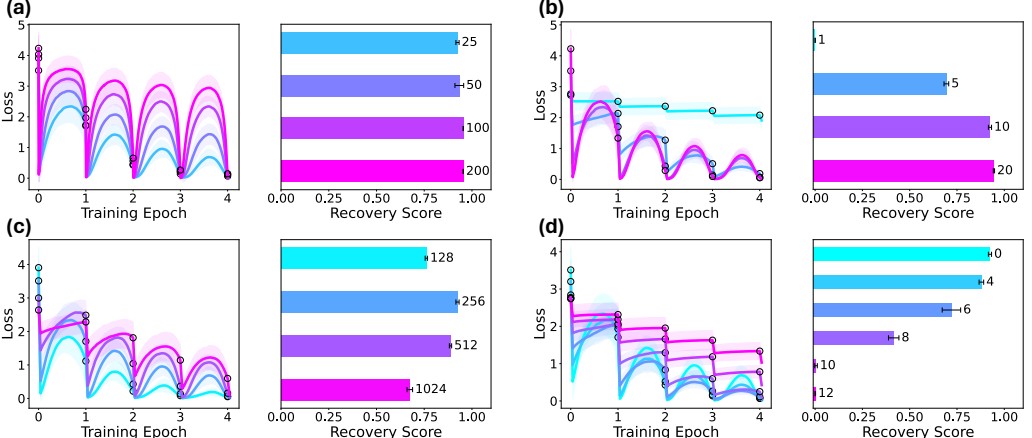

Figure 5: Effects of (a) number of documents (b) number of gradient steps (c) context length and (d) number of frozen blocks.

subsections we compute the recovery scores for different model sizes and training hyperparameters (Figures 2 and 5) to investigate their effects on the anticipatory recovery phenomenon.

## 3.2 Anticipatory Recovery is an Emergent Behavior

To study how the model size affects the amount of anticipatory recovery, we repeat this experiment with pre-trained Pythia models [11] of sizes 160M, 410M, 1.4B, and 2.8B. We plot the average loss curves as well as the recovery score for epoch 4 in Figure 2(a) . We observe that larger models clearly demonstrate stronger anticipatory recovery. The sharp increase of average recovery score from the 160M model to the 410M model indicates that anticipatory recovery is an emergent behavior.

**Anticipatory Recovery in Randomly Initialized Models.** To study whether anticipatory recovery is a result of pre-training, we repeat the experiments on randomly initialized models of different sizes, and plot the loss curves and average recovery scores in Figure 2(b). We follow the model initialization recipe of [11]. From the loss curves for the 410M and 1B models, especially in the last epoch, we see that the anticipation phenomenon also exists in randomly initialized LLMs. We observe that the anticipation effect is not as strong as in the pre-trained models. The effect of model size still holds: larger models clearly demonstrate stronger recovery.

**Effects of Model Width and Depth.** To further study the effect of model width and depth on the anticipatory recovery phenomenon beyond the model hyperparameters in the Pythia suite, we take a Pythia-1B model and vary the width (size of token embedding) and depth (number of transformer blocks) of the model and plot the average loss curves for cyclic training from random initializations in Figure 3. The original Pythia-1B model has token embedding of size 2048 and 16 transformer blocks. We observe that the model needs sufficient width (at least 512) and depth (at least 8 transformer blocks) to exhibit noticeable recovery, confirming that it is an emergent behavior contingent on model size.

## 3.3 Other Influential Factors

In this section we discuss the effect of other training hyperparameters on the anticipatory recovery phenomenon. We also include additional experiment details in Appendix A.1 and additional results in Appendix B.

**Number of Tasks.** Figure 5(a) plots the loss curves for different number of documents ($T \in \{10, 25, 50, 100, 200\}$). We observe clear recovery for all the curves, suggesting that the model can "memorize" a certain task transition even after training on 200 other tasks.

**Number of Gradient Steps.** Figure 5(b) plots training curves with different numbers of gradient steps taken on each document ($M \in \{1, 5, 10, 20\}$). More gradient steps in general leads to a higher recovery score, although in Appendix B.2 and B.8 we show that slight anticipation is still observed for 1 gradient step if we use a larger learning rate, and that the anticipation effect is stronger when the same total update is divided among more gradient steps by scaling the learning rate inversely with $M$.

**Context Length.** Figure 5(c) plots the loss curves for different context lengths ($C \in \{128, 256, 512, 1024\}$). Documents are padded to the same length with padding tokens if they are shorter than the specified context length. With the same number of gradient steps, larger context length is correlated with lower recovery score. This suggests that sufficient training on each task is necessary, and for longer input context it takes more gradient descent steps to memorize the task.

**Number of Frozen Blocks.** We experimented with freezing the first $B \in \{4, 6, 8, 10, 12\}$ transformer blocks in the pre-trained Pythia-1B model and tune only the last $16 - B$ blocks. Loss curves are plotted in Figure 5(d) . More frozen transformer blocks is correlated with lower recovery score. This observation is consistent with section 3.2 and confirm that the model needs sufficient depth to exhibit anticipatory recovery even with a frozen pre-trained deep representation.

**Optimizer.** In addition to the gradient descent optimizer, we experimented with the Adam optimizer [20]. Loss curves are plotted in Figure 6. We reset the optimizer state for each document. Results show that Adam, which is a stronger optimizer, further facilitates anticipatory recovery for both randomly initialized and pre-trained models.

**Data Randomness.** In realistic sequential learning setting the data points might be slightly different for each repetition (e.g. different descriptions of the same concept, different perspectives of the same object), leading to stochasticity in the optimization process. To explore sequential cyclic training with data randomness, we design the following two training settings: (1) we randomly mask a subset of the tokens in the input, and (2) we randomly shift the "window" of $C$ tokens used for training. The resulting loss curves are plotted in Figure 4. We observe that, while anticipatory recovery is generally weaker when there is more variation in each data point, the recovery still clearly exists.

**Summary.** The experiment results in this subsection suggest that the model's ability to fit on each task is crucial for the strength of anticipatory recovery. With a larger number of gradient steps, shorter context length, more learnable layers, and a better optimizer, the model is more capable of fitting to the focal task, and those factors also correlate with larger recovery score. We also confirmed that anticipatory recovery exists for long task sequences and slightly augmented data points within each episode, and again these factors that make learning harder also reduce anticipatory recovery.

## 3.4 Anticipatory Recovery in Vision Models

To examine the generality of the anticipatory recovery phenomenon, in this subsection we explore the sequential cyclic training setting on two tasks in computer vision: causal image modeling and image classification. More detailed experiment setup is available in Appendix A.

**Causal Image Modeling.** Similar to the LLM experiments, we fine-tune a pre-trained Image GPT model [14] on each sampled image from CIFAR-10 [18] for $M$ gradient steps, and repeat $E$ epochs with a fixed order of the images. The resulting loss curves are shown in Figure 7(a). The results show that the anticipatory recovery phenomenon also exists for sequential cyclic training of image modeling in addition to language modeling.

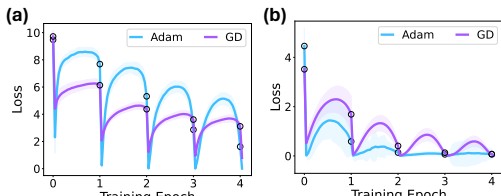
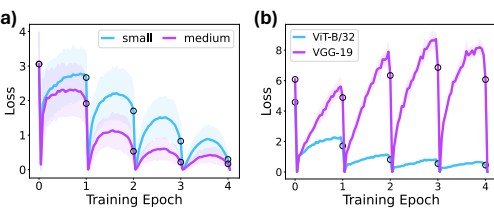

Figure 6: Comparison between Adam and vanilla gradient descent on (a) randomly initialized and (b) pretrained Pythia-1B models with cyclic training.

Figure 7: Results for cyclic training on (a) Causal image modeling with Image GPT. (b) image classification with vision transformers and VGG networks.

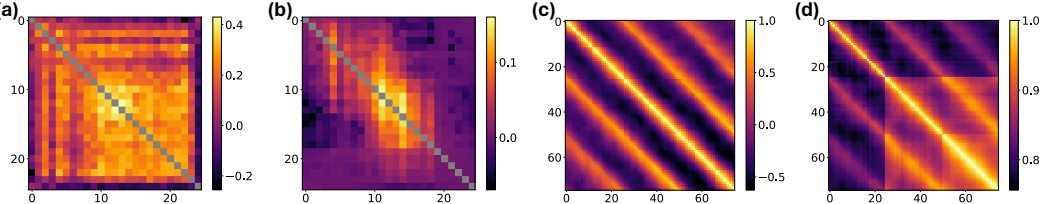

Figure 8: Heat map visualizations for (a) cosine similarities between the gradient vectors of the attention layer in transformer block 12 of the model for each task; (b) loss recoveries for training on task $x_i$ (y-axis) and evaluating on task $x_j$ (x-axis); (c) cosine similarities between the flattened model weight residuals at each point in training; (d) cosine similarities between the last layer activations for document $x_1$ at each point in training.

**Image Classification.** For each experiment, we randomly sample 800 images from Imagenet [19] and divide them into 25 batches of 32 images each. We fine-tune a pre-trained vision transformer (ViT) [15] and VGG-19 [16] model on each batch for $M$ gradient steps and repeat $E$ epochs with a fixed order of the batches. The resulting loss curves are plotted in Figure 7(b). Results show that both the transformer ViT and the convolutional VGG exhibit anticipatory recovery in cyclic training.

By these experiments we confirm that anticipatory recovery occurs not only for LLMs but also for at least some of the widespread image classification models and non-transformer architectures.

### 3.5 Online Loss Evaluation

We compare the performance of training with fixed ordering and random shuffling of each epoch in the prequential evaluation setting with $T = 100$ documents. Prequential evaluation [10] measures online performance by evaluating the model on the document it is about to be trained on, which is equivalent to evaluating the training loss of each batch. For the random shuffling condition, document 1 is always trained first in each epoch, and the other 99 documents are presented in a different random order in each epoch. In this experiment we set $C = 512$, $M = 10$, $E = 5$ and the error bars are based on 10 different seeds.

The resulting loss curves are plotted in Figure 1(c), and the average loss throughout each run is summarized in Table 1. We exclude epoch 1 when computing the average loss since all documents are new to the model for both the cyclic training and random shuffling condition. We observe that training with fixed ordering is superior to random shuffling in the prequential evaluation setting across all 4 pre-trained Pythia models of different sizes, due to the structure in the data stream. The results suggest practical potential of structured training.

## 4 Understanding Cyclic Training Dynamics

An important general question about anticipatory recovery is whether it is due to some causal mechanism relating the dynamics of model parameters to the training sequence, or whether it is more correlational in that adjacent tasks come to be represented more similarly by the model. We found initial evidence for the latter hypothesis in experiments locally reversing the task sequence (e.g., showing that $x_{t+1}$ primes $x_t$ nearly as much as vice versa). To further test this learned similarity hypothesis, we explore the relationships between tasks and the model's loss gradients, weights, and activations across

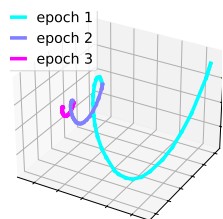

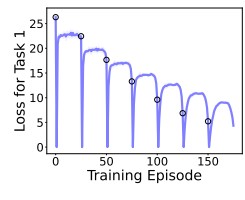

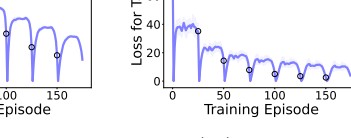

Figure 9: Top three PCA components of last layer weights in the first three epochs.

(a) $f_i(\boldsymbol{w}) = \boldsymbol{w}$

(b) $f_i(\boldsymbol{w}) = \boldsymbol{y}_i - \boldsymbol{w}$

Figure 10: Loss curve for task 1 in computational toy model, with different $f_i$. More experiment details in Appendix A.5.

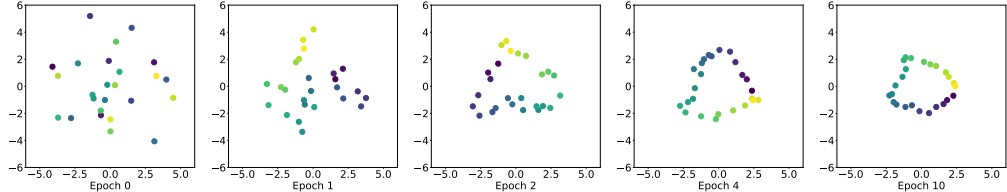

Figure 11: Visualization of PCA embeddings of the projected data points ($f_i^{-1}(\boldsymbol{P}\boldsymbol{x}_i)$, where $f_i(\boldsymbol{w}) = \boldsymbol{y}_i - \boldsymbol{w}$) in the toy model throughout training. Epoch 0 refers to the model before any training.

training history. The results enable us to better understand the dynamics of cyclic training. Unless otherwise stated, all visualizations in this section use the 410M model and default hyperparameters.

## 4.1 Temporal Structure of Gradients

We first explore the similarities of gradient information between documents during the training process. Our goal is to test the hypothesis that anticipatory recovery is mediated by increased similarity between gradients of proximal documents in our training sequence.

We do cyclic training for 4 epochs and compute the gradient of each document at the attention layer of transformer block 12 at the conclusion of training. In Figure 8(a), we plot the cosine similarities between these gradient vectors of each document. Results show that the gradients have mostly positive cosine similarities (except for the last document, on which the model has just been trained). To our surprise, the gradient similarities are highest near the center of the heat map rather than peaking along the diagonal. That is, the gradient similarity between documents $\boldsymbol{x}_{t-1}$ and $\boldsymbol{x}_t$ depends on where we are in the cycle. This result suggests an additional layer to the anticipatory recovery phenomenon: Recovery for document $\boldsymbol{x}_t$ is greatest from training on document $\boldsymbol{x}_{t-1}$, but the strength of the potential facilitation between $\boldsymbol{x}_{t-1}$ and $\boldsymbol{x}_t$ is actually greatest after we train for another $b$ documents (for some small number $b$). We verify this by computing the pairwise recovery: we take the model checkpoint after 4 epochs of cyclic training, and then for each pair of documents $(\boldsymbol{x}_i, \boldsymbol{x}_j)$, we do $M$ gradient updates on $\boldsymbol{x}_i$ and compute the difference in the loss of $\boldsymbol{x}_j$ before and after these gradient updates. We plot these pairwise loss recoveries in Figure 8(b). Results confirm that the amount of recovery on document $\boldsymbol{x}_j$ is highest when the model checkpoint is taken from roughly $b$ documents before or after document $\boldsymbol{x}_j$ in cyclic training and then fine-tuned on a proximal document $\boldsymbol{x}_i$ in the sequence. The fact that this pairwise loss recovery matrix is roughly symmetric also suggests that the anticipatory recovery phenomenon approximately exhibits task symmetry: gradient updates on document $\boldsymbol{x}_t$ decrease the loss for document $\boldsymbol{x}_{t+k}$ for small integers $k$, and vice versa. We provide additional visualizations for $T = 50, 100, 200$ in Appendix C and a more detailed description for this phenomenon.

## 4.2 Temporal Structure of Model Weights

We explore the structure of model weights along the optimization trajectory of cyclic training. We flatten and concatenate the model weight vectors after fine-tuning on each document. However, the cosine similarities between these raw model weight vectors are all very close to 1 without obvious structure, due to the proximity of model weights along the same optimization trajectory and numerical instability in huge weight vectors. To resolve these issues, we instead explore the

structure of "weight residuals." We compute the weight residuals by subtracting the average of weights in a window of length $T$ centered at the current document from the current weight, i.e. $w_{\text{res}}(t) = w(t) - \frac{1}{T}\sum_{n=t-T/2}^{t+T/2} w(n)$. This removes the shared components along the optimization trajectory and allows us to focus on the model weight updates for each document. Figure 8(c) visualizes a heat map of the cosine similarity between each pair of weight residuals from the second epoch to the fourth epoch. The visualization shows a cyclic structure in the weight residuals, as equidistant bright stripes that align with the training epochs. Furthermore, each stripe spans several documents, suggesting the similarity of weight residuals for proximal documents.

In addition to cosine similarities between weights, we explore visualizing the weights in a lower-dimensional space with Principle Component Analysis (PCA). We compute the top three PCs of the flattened last-layer weight vector (the output word embedding layer) for the Pythia-1B model, and plot its trajectory in Figure 9. The plot exhibits a clear helical structure that gradually converges. We believe this is highly relevant to anticipatory recovery: right before revisiting a task, the projected model weights in the helix move closer to the point corresponding to the previous appearance of that task, leading to anticipatory recovery on the loss of that task. As we go on with cyclic training, the model also exhibits less forgetting and gradually converges to a solution that achieves low loss on all tasks. It is important to note that the helical structure of the weight trajectory is not an obviously necessary consequence of the cyclical training. Cyclical training could be expected to yield a repeating pattern, but the facts that the tasks come to be organized in a circle that respects their ordering and that the trajectory goes through one full revolution per epoch (rather than some other arc length) are nontrivial and seem to be essential for anticipatory recovery.

### 4.3 Temporal Structure of Activations

In addition to gradients and weights, we visualize the trajectory of activations on a single document during the course of cyclic training. We do cyclic training for three epochs and save model checkpoints after fine-tuning on each document. We then compute the model activations before the output word embedding layer for document $x_1$ on each model checkpoint, and plot the cosine similarities between the flattened activation vectors in Figure 8(d). From the plot we can clearly observe the blocked pattern wherein the similarity between the activations become progressively higher across each epoch of cyclic training. This pattern suggests that every time we train on document $x_i$, the internal representation of $x_i$ in the model is more resistant to gradient updates on other documents $x_j$.

### 4.4 Computational Toy Model

To further understand the essential mechanism that yields anticipatory recovery, we design a minimalist "toy" simulation experiment. In this toy simulation, each task (formerly, document) $i \in \{1, \cdots, T\}$ is described by a single data point, $x_1, \cdots, x_T \in \mathbb{R}^N$. We assume a learnable linear embedding $P \in \mathbb{R}^{M \times N}$ that projects each $x_i$ into an $M$-dimensional embedding space. We also assume a learnable vector $w$ and task-specific mappings $f_i$, where $f_i(w)$ is the target for task $i$ in the same embedding space. We require each $f_i$ to be invertible as a simplifying assumption.

We define the loss for task $i$ as $\ell_i(P, w) = \frac{1}{2}\|Px_i - f_i(w)\|_2^2$. Just as when training a deep net, we assume here that representation learning occurs slowly, and that one training step for task $i$ involves a single gradient update of $P$ with step size $\alpha$:

$$P \leftarrow P - \alpha(Px_i - f_i(w))x_i^\top. \tag{1}$$

In contrast, at each training step, $w$, analogous to the fast-adapting weights in a neural network, can be rapidly tuned to solve for task $i$, yielding the loss minimizer conditional on $P$:

$$w \leftarrow f_i^{-1}(Px_i). \tag{2}$$

As in our main experiments, we sequentially optimize each $\ell_i$ as we iterate through the sequence of tasks. In each training step, we first update $P$ and then solve for $w$ given the update to $P$. Updating $P$ approximately reduces the distance between $Px_{i+1}$ and $f_{i+1}(f_i^{-1}(Px_i))$, which entails reducing the upper bound of $\|f_{i+1}^{-1}(Px_{i+1}) - f_i^{-1}(Px_i)\|$, assuming that each $f_i^{-1}$ is Lipschitz continuous. As a result of this optimization objective, the model will evolve along the optimization trajectory such that the $f_i^{-1}(Px_i)$ for all tasks $i$ gradually form a circular pattern. This gives an intuitive explanation

on the anticipatory recovery phenomenon, since updaing $\boldsymbol{w}$ according to equation 2 will also bring it closer to $f_{i+1}^{-1}(\boldsymbol{P}\boldsymbol{x}_{i+1})$, thus reducing the loss on task $i+1$ and exhibits anticipatory recovery.

We experimented with two very simple choices of $f_i$: $f_i(\boldsymbol{w}) = \boldsymbol{w}$ and $f_i(\boldsymbol{w}) = \boldsymbol{y}_i - \boldsymbol{w}$ for some task-dependent targets $\boldsymbol{y}_i$. We follow the same order over tasks—$1, \cdots, T$—for multiple epochs of training. The resulting loss curves are shown in Figure 10, which exhibits very similar anticipatory recovery trajectory as the full-blown LLM experiment. Visualizations of the 2-dimensional PCA embeddings for $f_i^{-1}(\boldsymbol{P}\boldsymbol{x}_i)$ in the second experiment are shown in Figure 11, which confirms our analysis that they gradually self-organize into a cyclic structure.

There are two potential reasons large overparameterized networks might produce the anticipatory recovery in a way analogous to the toy simulation. First, for larger networks, it is more likely that the network can develop task-specific parameters that quickly adapt to and memorize new input data, corresponding to Equation 2. And when the fast memorization is achieved, the gradient descent dynamics of the slow weights push the representations of the two adjacent tasks ($P\boldsymbol{x}_i$ and $P\boldsymbol{x}_{i+1}$) closer when $f$ is an identity function, according to Equation 1. This effect can be seen in earlier LLM experiments (Figure 2), where larger models achieve significantly lower losses within a few gradient update steps. Second, larger networks have more learning capacity to map the features of two adjacent tasks closer. In our linear projection model, anticipatory recovery keeps growing over many epochs, whereas the anticipatory effect is already at the strongest within 2 or 3 epochs in LLM experiments. Moreover, all data points are randomly generated in the toy model, which makes it easier to separate and map their representations according to a temporal structure than real-world data. In contrast, real-world data could require more representation capacity since data points are noisy and correlated.

### 4.5 Summary

In this section, we visualized model weight dynamics with heatmaps and we showed model activations and gradients during cyclic training. We discussed the special temporal structure that is exhibited in these heat maps. We also plotted the pairwise degree of recovery for fine-tuning on document $i$ and evaluating on document $j$, as well as the change of distance between fine-tuned model weights on different tasks. The results suggest that after we train on a document, the model's representation of that document becomes less sensitive to gradient updates on other documents. Finally, we showed a simple toy experiment that demonstrates a similar anticipatory recovery phenomenon in its loss curve, and discuss its connections to neural network training dynamics through the lens of task-specific and task-general parameters. Overall, these results shed some light on the dynamics of cyclic training.

## 5   Related Work

In this section we discuss the most relevant prior works to this paper. Please refer to Appendix D for additional related work.

**Cyclic and Structured Training.**   Prior theoretical works have studied convergence rates, under various assumptions, for the training setup where the data points are shuffled only once and that order is reused for all epochs [21, 22, 23, 24]. On the empirical side, [25] found that shuffling the data only once in the beginning can achieve a convergence rate comparable to shuffling every epoch. The training setup is equivalent to our cyclic training setup, but our research examines the loss on each task throughout the training cycle and discovers the anticipatory recovery effect. We also extend it to multiple gradient update steps on each data point.

**Online Learning.**   Online learning deals with the setting where the tasks come from an online sequential stream. One of the simplest algorithms in online learning is follow-the-leader [26], which stores all previous data from the stream and minimizes the total loss. It has strong performance guarantees but is computationally very expensive, and it also might not be feasible to store all the past data. Many subsequent works have developed cheaper algorithms under different assumptions [27, 28, 29]. Many recent works also explore the connection between online learning and meta-learning or continual learning [30, 31, 32, 33, 34, 35, 36]. The cyclic training setting that we explore in this research can be considered as a special case of the online learning setting where the data stream has a cyclic repetition structure. We employ multiple steps of online gradient descent [37] on each document from the stream and study the training dynamics of over-parameterized neural networks.

**Catastrophic Interference.** When transitioning between tasks sequentially, neural networks often experience "catastrophic interference" [9], marked by a significant drop in performance on previously learned tasks. Numerous algorithms have been proposed to mitigate catastrophic forgetting, focusing on general approaches including parameter regularization [38, 39, 40], data replay [41, 42, 43], knowledge distillation [44, 45, 46, 47], and architectural isolation and expansion [48, 49, 50, 51]. Our work extends interleaved training [52] to a larger number of tasks, specifically investigating the emergent anticipatory recovery phenomenon in cyclic training. This finding adds to the above literature by demonstrating a new mechanism by which large networks can avoid or recover from catastrophic interference.

**Continual Learning** Continual learning [8, 53, 54, 55] addresses a simplified setup where a model sequentially learns a set of tasks without revision. Recently, there have been debates over the practicality of continual learning setups. Studies like [56] have shown that as networks learn more tasks, they improve in learning speed and reduce forgetting. In large models, studies suggest that pre-trained vision classifiers can undertake continual learning with ease, by either freezing or fine-tuning representations [57, 58, 59]. In the language domain, research also suggests that LLMs exhibit emerging continual learning capabilities [60, 61]. Nevertheless, it is uncommon in real environments for tasks to occur only once yet for an agent to need to retain them. Unlike prior literature on continual learning, our research uniquely focuses on sequential learning environments with cyclic repetition.

## 6 Discussion and Limitations

In this work, we explored the training dynamics of overparametrized neural networks, especially LLMs, in sequential cyclic fine-tuning, where a finite set of documents are presented in the same order within each epoch. We demonstrated the remarkable phenomenon of anticipatory recovery—networks recover from the initial forgetting before seeing the same document again. The effect holds across many different network instances and training hyperparameters. This phenomenon is a sharp contrast with the well known phenomenon of catastrophic interference, where forgetting increases monotonically as a network is trained on a sequence of different documents.

We showed that anticipatory recovery occurs only when the network has sufficient width and depth and when it is well fitted to each document before moving to the next. Visualizations of model weights, model activations, and gradients exhibit clear temporal structure, which provide insights on the underlying mechanisms of anticipatory recovery.

Our research indicates that there is value in exploring naturalistic task sequences within continual learning, where tasks interleave in statistically regular patterns. This approach could expand the field's current focus on learning and retaining new tasks to also consider how effectively previously encountered tasks are re-learned when they reappear. With the anticipatory recovery phenomenon, we discovered a mechanism in which ML models can do surprisingly better than expected on prequential evaluation. By analyzing the different factors of model pre-training and fine-tuning that moderate this phenomenon, our experiments provide a promising first step toward leveraging structured training with agents in realistic environments.

**Limitations.** The cyclic training setup investigated in this work is distinct from the IID training setting assumed in the vast majority of the machine learning literature. It accounts for task repetition and task switching costs, which are critical components of the learning experience of humans and other real world agents. However, our current setup is still highly simplified. Future research could investigate the emerging training dynamics of neural networks in different types of structured environments, such as multiscale temporal dynamics [62], from both theoretical and empirical perspectives.

The mathematical foundation of anticipatory recovery also requires further investigation. Although our computational toy model reproduces the anticipatory recovery phenomenon, it does not explain why the effect is stronger in LLMs and autoregressive tasks than in other types of architecture or learning objectives. In terms of the empirical experiments, the ablation studies are only run in a single setting. Future research could run the experiments in more settings to reach more conclusive results and investigate possible alternative theoretical explanations to the anticipatory recovery phenomenon.

## Acknowledgment

We thank the Microsoft Accelerating Foundation Models Research program for providing Azure cloud compute credits. We thank members of the NYU Agentic Learning AI Lab for helpful discussions. The compute was supported by the NYU High Performance Computing resources, services, and staff expertise. MJ was supported by NSF grant 2020-906.

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

# A    Additional Experiment Details

## A.1    LLM Experiments

We use the Huggingface Transformers Library [63] for fine-tuning the LLMs. The learning rate 0.001 for vanilla gradient descent and 0.00001 for Adam. For all experiments we run 3 to 5 trials with different random seeds, except the results in Figure 1 which are based on 20 seeds. The shaded area in the figures denotes standard deviation among trials and documents (for shift-averaged loss curves).

## A.2    Causal Image Modeling Experiments

**Models**    Image GPT [14] is a GPT-2-like model trained to predict the next pixel value in an image. It is pre-trained on the Imagenet dataset [19] resized to 32x32. The Image GPT authors provide three pre-trained models of different sizes. In our experiments, we use the Image GPT-small and Image GPT-medium models.

**Datasets**    We use the CIFAR-10 [18] dataset for fine-tuning. For tokenization, the pixel RGB values are categorized into 512 pre-determined clusters with the nearest-neighbor classifier, as in [14]. After pre-processing, each image is transformed to a sequence of length 1024, with code book of size 512.

**Training Setup**    We did not manage to sequentially fine-tune the model stably with the dropout layers, so the dropout layers are turned off during the Image GPT fine-tuning experiments. We use the Adam optimizer [20] with learning rate 0.001. The default hyperparameters in the experiments are $T = 25$ images, $M = 10$ gradient update steps, $E = 5$ epochs. Same as the LLM experiments, we use the average cross-entropy loss as our evaluation metric.

## A.3    Image Classification Experiments

The images are resized to 256x256 followed by a center crop of 224x224. We use the Adam optimizer with learning rate 0.0001 for $M = 10$ gradient steps on each batch of images.

## A.4    Shift-averaged Loss Calculation

The shift-averaged loss curves plotted in Figure 1(b) are calculated by replicating Figure 1(a) on each document in the training sequence, re-aligning these curves so that 0 on the x-axis always represents the moment before the first occurrence of the focal document, and average them. For example, if the length of the sequence is 50, then for training epoch 0.5 on the x-axis, we take the loss of document 1 after training on document 25; the loss of document 2 after training on document 26; ...; the loss of document 50 after training on document 24 of the next epoch; and average these losses. Subsequent figures (Figure 2-7, 12-19) are plotted with the same approach.

## A.5    Computational Toy Model

For Figure 9a, we pick $f_i(\boldsymbol{w}) = \boldsymbol{w}$, and each data point $\boldsymbol{x}_i$ and $\boldsymbol{w}$ is a vector of length $N = M = 1000$. We have $T = 25$ data points and use the vanilla gradient descent optimizer with learning rate 0.01. The projection matrix is initialized with every entry sampled independently from $\mathcal{N}(0, 1/N^2)$. Each entry of the data points $\boldsymbol{x}_i$ and $\boldsymbol{w}$ is sampled independently from $\mathrm{Unif}(-1, 1)$. For Figure 9b, we pick $f_i(\boldsymbol{w}) = \boldsymbol{y}_i - \boldsymbol{w}$, $N = M = 100$, $T = 25$, and learning rate 0.01. Each entry of $\boldsymbol{y}_i$ is also sampled independently from $\mathrm{Unif}(-1, 1)$.

## A.6    Compute Resources

Each experiment presented in the paper is run with one NVIDIA A100 GPU, 2 CPUs, and 32GB of RAM. The training time highly depends on the hyperparameter choices, especially model size and number of gradient steps. The longest fine-tuning experiment with 20 gradient steps per episode on the Pythia-1B model takes roughly 30 minutes under this setup. The minimal compute resource requirement needed to reproduce the experiments with a Pythia-1B model is one GPU with 16GB of GPU memory.

# B    Additional Experiment Results

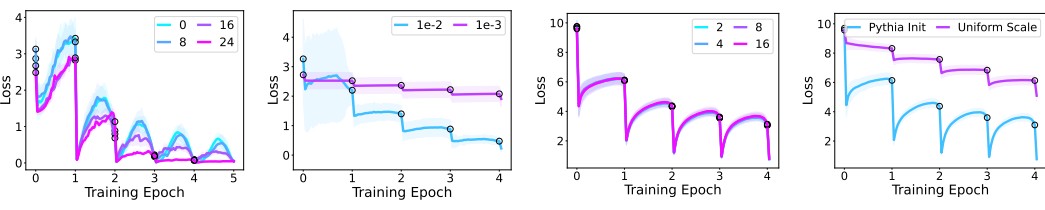

Figure 12: Effect of partial document shuffling.

Figure 13: Effect of learning rate in 1-step GD.

Figure 14: Effect of number of attention heads.

Figure 15: Effect of model initialization.

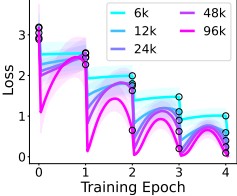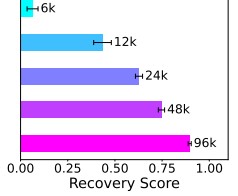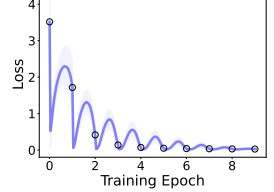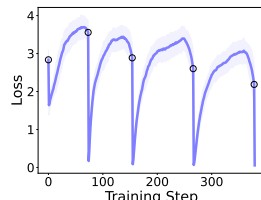

Figure 16: (Left) Effect of pre-training steps. The full pre-training process is 143k steps. (Right) Recovery scores for models with different pre-training steps.

Figure 17: Loss curve for cosine learning rate schedule.

Figure 18: Effect of inserting random documents after the repeating sequence.

## B.1    Partial Random Shuffling

Throughout the paper we have been focusing on the setting where the document ordering is sampled once and stay fixed for all epochs. What if we only fix the first and last document, and shuffle the documents in between? We experimented with shuffling the documents from document $x_2$ through $x_N$ for $N \in \{8, 16, 24\}$ every epoch. In Figure 12 we plot the loss curves for document $x_1$. From the loss curves we can observe that even when $N = 24$ we can still observe some anticipatory recovery, suggesting that the order of the tasks between two consecutive repetitions of the $x_{25}$ and $x_1$ can be arbitrary for us to observe recovery on $x_1$.

## B.2    One-step Gradient Descent with Larger Learning Rate

In Figure 5(b) we observe that there is no anticipation when we take only one gradient descent step on each document with learning rate 0.001. We experimented with one-step gradient descent using a higher learning rate, 0.01. We plot the resulting average loss curves under the same training setup in Figure 13. We observe that, with a larger learning rate, slight anticipation is still observed for 1 gradient step.

## B.3    Effect of Number of Attention Heads

In addition to varying the model width and model depth in Figure 3, we also experimented with varying the number of attention heads $h \in \{2, 4, 8, 16\}$ while keeping model width to be 2048 and model depth to be 16. Loss curves on document $x_1$ are shown in Figure 14. The results suggest that the number of attention heads does not have a big effect on cyclic training in our setting.

## B.4    Effect of LLM Model Initialization

Here we compare the performance of the initialization scheme used by [11] (also used for all randomly initialized models in the main text) and a simple initialization scheme that samples the weight matrices from an isotropic Gaussian distribution with $\sigma = 0.02$. The loss curves for document 1 under these two initializations of the Pythia-1B model are plotted in Figure 15. We observe that Pythia's initialization scheme achieves much better average loss and also exhibits stronger anticipatory recovery. This demonstrates the importance of LLM initializations. The result is consistent with our

observations in section 3.3 that the model's ability to fit on each task is correlated with the amount of anticipatory recovery.

## B.5  Effect of Pre-training Steps

In addition to comparing pre-trained models and randomly initialized models in Figure 2, we further study the effect of model pre-training by examining model checkpoints with different numbers of pre-training steps. We took Pythia models pre-trained for 6K, 12K, 24K, 48K, and 96K steps and plot the shift-averaged loss curves for cyclic fine-tuning in Figure 16. We found that more pre-training does give rise to higher anticipatory recovery. As we summarize at the end of section 3.3, we hypothesize this result fits a broader pattern in which the strength of the anticipatory recovery effect is related to how well the model can fit each successive training task. Models with more pre-training steps are more capable of fitting each successive training task, and therefore exhibit higher anticipatory recovery.

## B.6  Effect of Cosine Learning Rate Schedule

For experiments in the main paper we used a constant learning rate during the fine-tuning process. To study whether anticipatory recovery occurs in typical LLM optimization schemes, we experimented with cosine learning rate scheduling on the Pythia-1B model with 10 epochs (minimum learning rate = 0, maximum number of epochs = 10), and plot the results in Figure 17. We show that the model also exhibits the anticipatory recovery effect in other learning rate schedules.

## B.7  Effect of Inserting Random Documents after the Repeating Sequence

To examine how the anticipatory recovery effect may generalize to other forms of structured training, we experimented with a new setting where only the first 20 documents are kept fixed in each epoch, and a random number of other documents (between 20 and 100) are inserted after the first 20. These random padding documents appear only once in the entire sequence. This new setting generalizes cyclic training in that (1) rather than having the same documents in every epoch, we insert random other documents between every repetition (2) epochs can have different lengths. The resulting loss curve is plotted in Figure 18. We still observe anticipatory recovery for documents 2 through 20 in this setting, suggesting that anticipatory recovery exists as long as there is a repeating sub-sequence in the data stream.

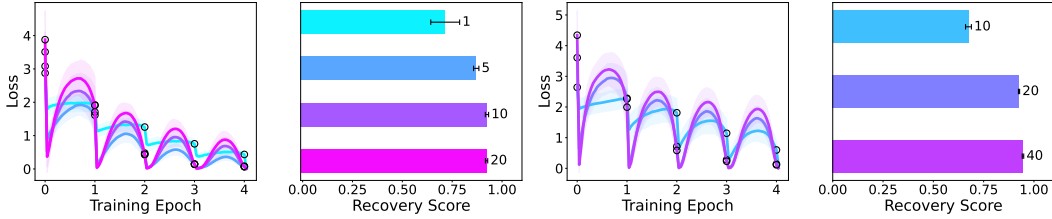

(a) Number of Gradient Steps with Inverse LR Scaling   (b) Number of Gradient Steps for Context Length 1024

Figure 19: Effect of number of gradient steps (a) with inverse learning rate scaling and (b) for context length 1024.

## B.8  Effect of Number of Gradient Steps with Inverse Learning Rate Scaling

In Figure 19a we experimented with inversely scaling the learning rate with the number of gradient steps. We use a learning rate of $0.01$ for $M = 1$, learning rate $0.002$ for $M = 5$, learning rate $0.001$ for $M = 10$, and learning rate $0.0005$ for $M = 20$. The results suggest that the anticipation effect is stronger when the same total update is divided among more gradient steps.

## B.9  Effect of Number of Gradient Steps for Long Context Length

In Figure 19b we experimented with different number of gradient steps $M \in \{10, 20, 40\}$ for context length 1024. The results confirm that longer context length is not a fundamental limitation to

anticipatory recovery, and we can achieve the same recovery score as a smaller context length with more gradient steps.

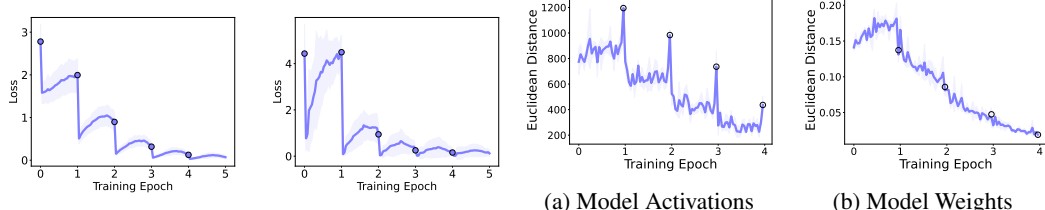

Figure 20: Experiments with GPT2-large.

Figure 21: Experiments with wikitext-103 dataset.

(a) Model Activations

(b) Model Weights

Figure 22: Magnitude of (a) model activation updates and (b) model weight updates through cyclic training.

## B.10 Experiments with GPT-2

To evaluate how the anticipatory recovery phenomenon generalizes across different LLM architectures, we experimented with GPT-2 architecture [64], specifically the GPT2-large pre-trained model (812M parameters) on the CNN/Daily Mail dataset. The loss curve for document 1 is plotted in Figure 20. The model consistently observed anticipatory recovery. Note that GPT-2 is the predecessor of many modern LLMs and therefore the results further suggest that the anticipatory recovery phenomenon is prevalent among more recent LLM architectures.

## B.11 Experiments with Wikitext

To evaluate how the anticipatory recovery phenomenon generalizes across different natural language datasets, we experimented with the wikitext-103 dataset [65], which contains over 100 million tokens from articles on Wikipedia. Since Wikipedia data is part of the pre-training dataset of Pythia, we only experiment with randomly initialized models. The loss curve for document 1 is plotted in Figure 21. The result suggest that the anticipatory recovery phenomenon is generalizable to different data sources.

# C Additional Visualizations

## C.1 Magnitude of Changes in Model Weights and Model Activations

We plot the magnitude of the difference between the $x_1$ activations of model checkpoints saved at consecutive episodes throughout four epochs of cyclic training of a Pythia-410M model in Figure 21a, and observe a clear stepwise pattern. In contrast, the magnitude of model weight updates (Figure 21b) decreases monotonically over the training episodes and do not exhibit this stepwise pattern. This result is consistent with the pattern we observe in section 4.3.

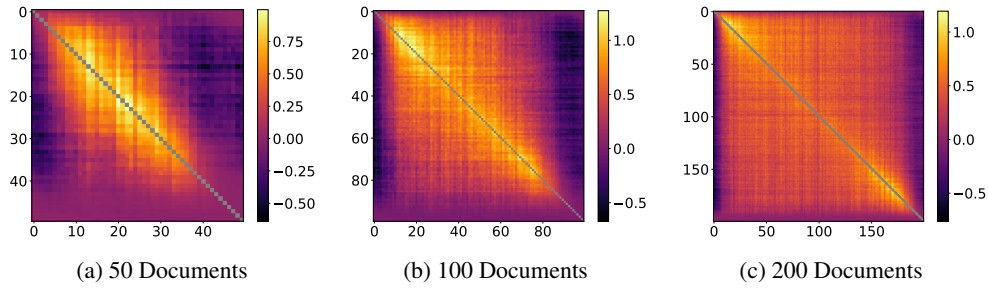

(a) 50 Documents

(b) 100 Documents

(c) 200 Documents

Figure 23: Loss recoveries for training on task $x_i$ (y-axis) and evaluating on task $x_j$ (x-axis) for longer document sequences of different lengths.

## C.2 Pairwise Recovery Matrices for Longer Document Sequences

Similar to Figure 8(b), we plot the pairwise loss recoveries for each pair of documents $(\boldsymbol{x}_i, \boldsymbol{x}_j)$ in longer document sequences, where $T = 50, 100, 200$ respectively, in Figure 23. We use the 1B model and default hyperparameters. We observe that, as we increase the length of the document sequence, the highlight area near the center of the matrix is separated into two blobs, one in the top-left corner and the other in the bottom-right corner. We also observe a "boundary" on the sides of the matrix where there is little or no recovery. The width of this "boundary" stays relatively constant across different lengths of document sequences and is around 10 to 15 documents. This confirms our observation in the main text that the recovery on document $\boldsymbol{x}_j$ when fine-tuning on a proximal document $\boldsymbol{x}_i$ is highest when the model checkpoint is taken from document $\boldsymbol{x}_{j \pm b}$ where $b$ is a small number relative to the length of the document sequence.

# D  Additional Related Work

**Learning in Structured Environments.**  Our research also relates to the more general topic of learning in structured environments. [62] studied regression and classification tasks with multi-scale temporal structure in the environment characterized by $1/f$ dynamics. While the cyclic training setting that we study is a more simplified setup than that of [62], we aim at unveiling more insights on applying standard SGD on over-parameterized networks. A potential direction for future work would be to study anticipatory recovery in regimes with richer, hierarchical sequence structure.

**LLM Emergent Capabilities.**  Recent advancements in large-scale Transformer networks [66, 1] have demonstrated exceptional ability to model long sequence language data. Beyond basic language modeling and downstream task performance, these models have shown emergent behaviors [67] that appear to manifest only beyond a certain model scale [2, 68, 69, 70]. Related to our research, recent studies reveal that LLMs possess remarkable memorization skills, enabling them to recall news sentences after just a few exposures [11, 71, 72]. However, the sequential learning dynamics behind such memorization have not been thoroughly examined. Our work comprehensively explore the sequential learning setting with cyclic task repetition and demonstrates task anticipation, a new emergent capability of large models.

# E  Broader Impact

This research deepens our understanding of catastrophic interference in naturalistic training setups. This understanding could lead to the design of better training algorithms of LLMs and other large neural networks that are more similar to human learning. These algorithms may give rise to more powerful and embodied AI systems with online adaptive learning capability, which may have many potential societal consequences.

Recovery from catastrophic interference might be undesirable in scenarios where some documents are intended to be forgotten. Our research provide more understanding into the anticipatory recovery phenomenon. Future research into the unlearning problem as well as a deeper understanding of the training dynamics of different types of structured environments can help further address the privacy issue.

