# OpenReview forum: "Reawakening knowledge: Anticipatory recovery from catastrophic interference via structured training"
_NeurIPS.cc/2024/Conference — NeurIPS 2024 poster_

### Official Review · Reviewer_5Cpf · 2024-07-03

**Soundness:** 3
**Presentation:** 3
**Contribution:** 2
**Rating:** 4
**Confidence:** 4

**Summary:**

This paper explores the training dynamics of neural networks, particularly language models in a structured training setting where examples are presented cyclically. The paper discovered a phenomenon called "anticipatory recovery," where models in each training epoch recover from forgetting of an example right before encountering the example. The paper show that the behavior is due to the temporal structure of gradients and model weights in cyclic training, and provided a theoretical characterization of the phenomenon using a toy model.

**Strengths:**

- The phenomenon of "anticipatory recovery" during cyclic training is a novel discovery and contrasts with the well-known issue of catastrophic forgetting.
- The experiment is comprehensive, examining not just language models but also vision models and other architectures.
- The analysis part is clear and well-structured, with detailed visualizations and theoretical analysis on a toy model to support and explain the empirical findings.

**Weaknesses:**

The main weakness of the paper is its unclear connection with common practices in training neural networks, therefore, it can be a little bit difficult to appreciate the significance of the findings.

  - The paper contrasts the anticipatory recovery phenomenon with catastrophic forgetting, but this is an unfair comparison as anticipatory recovery happens only when the model is trained on the same data repeatedly, while catastrophic forgetting mostly happens when the model is trained on new data and never again have access to the old data. There is typically no catastrophic forgetting in multi-epoch training setting, and anticipatory recovery cannot happen if the old data is never seen again.

  - The anticipatory recovery phenomenon may have limited practical implications, for 1) training with fixed example order usually achieves worse generalization performance than random-shuffling due to lack of mini-batch gradient diversity. 2) although anticipatory recovery relieves forgetting, it only reduces the forgetting of the next few examples in the coming sequence, rather than reducing forgetting of all previous examples as is the goal in continual learning. 3) the model weight seems to be forming a cyclic structure under cyclic training, which pushes the model weight to become closer to its corresponding state in previous epochs. Would this also mean that the model is farther apart from its state in the other half of the epoch (compared to random-shuffle training)? If so, less forgetting of the next few examples may come at the cost of more forgetting of the other examples in the dataset.

  - The author proposes to examine the potential of structured training, but it is not clear how the anticipatory recovery phenomenon can benefit structured training. Why is recovery from forgetting right before training useful, when the model is soon to be trained anyway? Also, the anticipatory recovery phenomenon seems only exist in the cyclic training setting, which is a very special form of structured training. It is not clear how the findings can be generalized to other forms of structured training.

**Questions:**

Can you explain why the prequential evaluation setting is relevant, and why lower loss in prequential evaluation implies potential of structured training?

Can you explain why the particular toy model is chosen in the theoretical analysis? Is there an intuitive explanation for the loss function in the toy model?

**Limitations:**

The authors adequately addressed the limitations.

---

> ### Author Rebuttal · Authors · 2024-08-07
>
> Thank you for the detailed feedback.
>
> **Re: Unfair comparison for catastrophic forgetting and anticipatory recovery.**
> > We agree with the reviewer that there is no catastrophic forgetting if all tasks are trained in every epoch, or if training passes over all tasks many times. However, in our work we show that the forgetting and recovery can be observed within each training cycle, before the same document is encountered again. This is a different timescale than the reduction of forgetting in typical multi-epoch training, and is a surprising phenomenon. Prior to our paper, the accepted wisdom in the continual learning community was that catastrophic forgetting would arise within each cycle as we train on other tasks before training on the focal task again. We show an important case where the catastrophic forgetting effect is largely mitigated due to the anticipatory recovery towards the end of the training cycle, which challenges the accepted wisdom.
>
> **Re: “The anticipatory recovery phenomenon may have limited practical implications.”**
> > We agree with the reviewer that for cyclic training, compared to random shuffled training, less forgetting of the next few examples comes at the cost of more forgetting of the other examples in the dataset. However, we argue that what real-world agents care about is the performance on tasks they actually encounter at the times they encounter them. If task order is (approximately) repeating, then a bias toward good performance on upcoming tasks at the cost of other tasks is a good thing, as shown in figure 1c of the paper.
>
> **Re: “Why the prequential evaluation setting is relevant, and why lower loss in prequential evaluation implies potential of structured training.”**
> > As we explain in the general rebuttal, we introduce cyclic training not as a method but as an objective. Because many natural environments are quasi-cyclic, it is important to evaluate existing methods in this setting, as opposed to traditional CL setups. For an agent that acts and learns in the real world, the most important metric it cares about is its performance on the tasks it actually encounters, at the times it encounters them. This justifies the relevance of prequential evaluation in building and evaluating such agents. According to the prequential view, training and testing are two sides of the same coin. While it is correct that the model is about to be trained on the next task, its loss on the next task is determined by how good its predictions are prior to the update. Lower loss in prequential evaluation suggests that certain models can automatically enjoy an anticipation benefit when trained on structured sequences (where task sequences tend to repeat) and catastrophic forgetting might be less of an issue for these models in these structured environments.
>
> **Re: “It is not clear how the findings can be generalized to other forms of structured training.”**
> > In addition to pure cyclic training, we also experimented with some variants in the paper, including data randomization (Figure 4, lines 148-154) and partial random shuffling (Figure 12, lines 602-608), and demonstrated anticipatory recovery in these cases. In rebuttal experiment E3, we experimented with another variant where only the first 20 documents are kept fixed in each epoch, and a random number of other documents are inserted in between. This new setting generalizes cyclic training in that (1) rather than having the same documents in every epoch, we insert random other documents between every repetition (2) epochs can have different lengths. We still observe anticipatory recovery for documents 2 through 20 in this setting, suggesting that anticipatory recovery exists as long as there is a repeating sub-sequence in the data stream.
>
> **Re: “Why is the particular toy model chosen in the theoretical analysis.”**
> > The particular toy model is chosen for theoretical analysis because it is a minimalist setting where we can reproduce the anticipatory recovery phenomenon and each component in the toy model has a natural and intuitive counterpart in the more complex LLM training procedure. The L2 loss function is chosen due to its wide usage in machine learning applications. We believe that similar analysis can be done for other common loss functions such as the cross entropy loss as well.

---

> > ### Comment · Reviewer_5Cpf · 2024-08-10
> >
> > Thank you for your response to my questions. While I appreciate the authors' effort put into the work and the response which provides more discussion around the limitations of the work, these fundamental limitation still exists. Therefore, I prefer to keep the current score.

---

### Official Review · Reviewer_gu9E · 2024-07-12

**Soundness:** 3
**Presentation:** 2
**Contribution:** 3
**Rating:** 7
**Confidence:** 2

**Summary:**

The authors identify a phenomenon termed anticipatory recovery, where if the model is trained on a fixed shuffled data sequence for multiple epochs, the loss of a training sample will first increase and then decrease again right before it is met again. Extensive experiments have been conducted to show the phenomenon happens across different model size, optimization, initialization states, and several hyperparameters etc. The authors also show the existence of certain cyclic structure in data gradient, attention, and activations. Finally, a simple toy example is demonstrated for intuition.

**Strengths:**

The phenomenon is very interesting, and the experiment settings are well considered and extensive.

**Weaknesses:**

In terms of presentation, several settings and contributions can be improved by providing more discussion. The analysis of the computational toy model can be improved.  See **questions** for more detail.

**Questions:**

1. In Figure 1(b), how exactly is the loss computed? Say when you plot the figure for the 0.5th epoch, how is the number obtained over all documents?
2. Figure 5 needs legends.
3. Can you give more context on the prequential evaluation? What I had in mind about prequential coding is the same as looking at the training loss of each batch (see [1]).
4. For figure 8(a, b), as a sanity check, if you reshuffle the data sequence (and fix that sequence for cyclic training), the same structure will show, is that correct?
5. Around line 273-275, is the argument theoretically justified?
6. From figure 6(a), we see that the anticipatory recovery happens with Adam but not quite with GD if the model is randomly initialized. Intuitively, Adam keeps a memory of the data sequence, but not GD. However, when the model is pretrained, then both GD and Adam show anticipatory recovery. This seems to suggest that pretraining is one of the most important factors. I would like to hear more from the authors.
7. Can you explain more about the significance of Figure 8(c)? What is the $T$ used for this figure? The residual is the averaged weight residual over $T$ steps, so for each $t$, the residual will have $T-1$ overlaps, doesn't the overlap contribute significantly to the cyclic structure?

[1] The Description Length of Deep Learning Models

**Limitations:**

Limitations have been discussed.

---

> ### Author Rebuttal · Authors · 2024-08-07
>
> Thank you for the detailed feedback.
>
> **Re: “In Figure 1(b), how exactly is the loss computed?”**
> > The loss in figure 1(b) is computed by replicating figure 1(a) on each document in the training sequence, and re-aligning these curves so that 0 on the x-axis always represents the moment before the first occurrence of the focal document. For example, if the length of the sequence is 50, then for the 0.5th epoch, we take the loss of document 1 after training on document 25; the loss of document 2 after training on document 26; …; the loss of document 50 after training on document 24 of the next epoch; and average these losses.
>
> **Re: “Figure 5 needs legends.”**
> > The legends in figure 5 are in the bar plots. For example, in figure 5(a), the light blue curve in the left subfigure corresponds to the light blue bar in the right subfigure, which shows the loss curve for 25 documents.
>
> **Re: “Can you give more context on the prequential evaluation?”**
> > Yes, prequential evaluation is the same as looking at the training loss of each batch. We argue that the performance on the upcoming task is what matters for an agent which continuously acts and learns in the real world, and prequential evaluation measures this performance. We will explain this better in the revision.
>
> **Re: “For figure 8(a, b), if you reshuffle the data sequence, the same structure will show.”**
> > Yes, the data sequence is randomly selected at the beginning and fixed for cyclic training. The same structure will show for any random data sequence, provided it is the same on every cycle.
>
> **Re: “Around line 273-275, is the argument theoretically justified?”**
> > Let $z_1 = Px_{i+1}$ and $z_2 = f_{i+1}(f_i^{-1}(Px_{i}))$, and assume that $f_{i+1}^{-1}$ is Lipschitz continuous. By the definition of Lipschitz continuity, we have  $f_{i+1}^{-1}(Px_{i+1}) - f_i^{-1}(Px_{i}) = |f(z_1)-f(z_2)| \leq K|z_{1}-z_{2}|$ for some positive real constant K, so when we reduce $|z_{1}-z_{2}|$ we also reduce the upper bound for $f_{i+1}^{-1}(Px_{i+1}) - f_i^{-1}(Px_{i})$. Although this argument does not directly prove that  $f_{i+1}^{-1}(Px_{i+1}) - f_i^{-1}(Px_{i})$ will necessarily decrease, we think that proving the decrease of its upper bound serves as sufficient justification for this computational toy model.
>
> **Re: “This seems to suggest that pretraining is one of the most important factors.”**
> > Thanks for the good suggestion. In fact, GD exhibits anticipatory recovery without pre-training as well, especially in the later epochs. That said, more pre-training does make anticipatory recovery more significant. In rebuttal experiment E1, we took Pythia models pre-trained for a different number of steps (6K, 12K, 24K, 48K, 96K), and we found that more pre-training does give rise to higher anticipatory recovery (see Figure 1 of the rebuttal PDF).  We hypothesize this result fits a broader pattern in which the strength of the anticipatory recovery effect is related to how well the model can fit each successive training task (see section 3.3). Models with more pre-training steps are more capable of fitting each successive training task, and therefore exhibit higher anticipatory recovery. We plan to further study the effect of pre-training in future work.
>
> **Re: “Can you explain more about the significance of Figure 8(c)?”**
> > For this figure we used $T=25$. We want to note that, as seen from the equidistant stripes with interval $T$ in figure 8(c), the weight residuals at step $i$ have high similarity with those of $i+T$ and $i+2T$, while there is no overlap in their averaging window. For example, the weight residuals from step 25 and step 50 are highly correlated (with cosine similarity close to 1), but their averaging windows have no overlap at all; on the other hand, the averaging windows of step 25 and step 35 have roughly 50% overlap but the cosine similarity of the weight residuals at step 25 and step 35 is negative. Hence the cyclic structure is not due to overlaps in the averaging window. The significance of figure 8(c) is that, along with figure 9, it reveals the temporal structure of model weights in the cyclic training process.

---

> > ### Comment · Reviewer_gu9E · 2024-08-12
> >
> > Thanks for the detailed response and my questions are addressed. I have some minor comments (that do not affect my ratings)
> > 1. I suggest the authors explain the figure 1(b) loss calculation in the revision as well (could be in the appendix).
> > 2. In figure 8, personally I would put the recovery score to the right of the bar, and have a separate legend, this is not crucial.
> > 3. There also exists some work that relates generalization and compression (via prequential coding), which can be mentioned in the related works section, and they are also good references to why prequential coding is relevant, see [1, 2].
> >
> > [1] Sequential Learning of Neural Networks for Prequential MDL
> > [2] Continual Learning from the Perspective of Compression

---

> > > ### Author Response · Authors · 2024-08-13
> > >
> > > We thank the reviewer for the additional valuable feedback. We will address these comments in the revision.

---

### Official Review · Reviewer_4gi1 · 2024-07-16

**Soundness:** 3
**Presentation:** 3
**Contribution:** 1
**Rating:** 5
**Confidence:** 4

**Summary:**

The paper studies a specific kind of structured training dubbed cyclic training, where documents (or tasks) are presented cyclically in a fixed, repeated sequence. The paper reveals that LLMs exhibit anticipatory recovery behavior, i.e. they begin to recover from forgetting a document before encountering it again in the sequence. The paper offers some insights into the training dynamics in the cyclic training setup. emphasizing that this anticipatory behavior is not exclusive to LLMs but can be observed in other large-scale networks as well. The paper investigates the reasons for this anticipatory recovery, finding initial evidence that suggests adjacent tasks in the training sequence are represented more similarly by the model.

**Strengths:**

1. The paper identifies a surprising phenomenon where models exhibit anticipatory recovery, which is counter-intuitive given the understanding of how LLMs handle sequential knowledge without explicit memory mechanisms
2. It presents extensive experiments demonstrating that anticipatory recovery occurs not only in LLMs but also in other large-scale vision models. It shows that this behavior persists across several pretrained Pythia large models and, to a lesser extent, in randomly initialized models. The study further examines various factors, such as the number of tasks, training parameters, and data randomness, providing robust evidence for the phenomenon.
3. The paper provides insights into the relationship between the anticipatory recovery phenomenon and the training dynamics.

**Weaknesses:**

1. While the anticipatory recovery phenomenon is intriguing and counterintuitive, its practical value remains unclear. The paper investigates implications on Online Loss Evaluation, but the experiments are limited and do not convincingly demonstrate why cyclic training would perform better than random training. It is also uncertain if the observed benefits are directly tied to anticipatory recovery or influenced by other experimental confounders.
2. The motivation for using cyclic training is not particularly convincing. Cyclic training appears less natural compared to continual learning, and it is unclear if understanding training dynamics in this setup provides deep insights into the training of large models. This connection is weak and requires further development to establish its significance.
3. Although the experiments are thorough, they are somewhat narrow in scope. The study primarily focuses on Pythia pretrained models and evaluates them using the CNN/Daily Mail dataset. It would have been beneficial to include a broader range of models and evaluation datasets. Additionally, the distinction between context length in the training setup and the models' context length is not clearly explained, leaving ambiguity about its impact on loss evaluation.
4. While lesser priority, the paper lacks citations for a substantial body of work on continual learning in LLMs, such as AdapterCL by Madotto et al. (2020), Lfpt5 by Qin & Joty (2021), and ProgPrompt by Razdaibiedina et al. (2023).

**Questions:**

Can you please address each of the weaknesses presented above?

---

> ### Author Rebuttal · Authors · 2024-08-07
>
> Thank you for the detailed feedback.
>
> **Re: “its practical value remains unclear.” and “cyclic training appears less natural compared to continual learning”.**
> > We would like to argue the opposite, that traditional continual learning is less natural than cyclic training. Most standard benchmarks and methods in continual learning (CL) assume IID task sampling, or every task trained only once; and the objective is to perform well on all tasks under these assumptions. Both the assumptions and the objective are unnatural: tasks naturally repeat in day-to-day life, with predictable transitions (e.g., a person might always eat breakfast after brushing teeth) and the main metric that an agent which acts and learns in the real world cares about is its performance on the tasks it actually encounters, at the times it encounters them (which justifies the prequential evaluation experiment in the paper, Table 1). As a result, cyclic training is closer to the challenges facing real-world agents than standard class- or task-incremental continual learning.
>
> > In this paper, we are not trying to argue that cyclic training performs better than random training in terms of average performance on all tasks. Instead, we argue that, when training machine learning models on naturalistic sequences (where task sequences tend to repeat), certain models automatically enjoy an anticipation benefit. The experiments in our paper aim at identifying what factors influence the anticipatory recovery effect, so that we can use the right models and optimizers to maximize this effect.
>
> **Re: “[the experiments] are somewhat narrow in scope”.**
> > While many of the experiments are conducted on Pythia models and the CNN/Daily Mail dataset to identify the influential factors of the anticipatory recovery effect, we also carried out many experiments with other models and datasets, including vision models (Figure 7, Section 3.4), the GPT-2-Large model (Figure 17, Appendix B.7), and wikitext-103 dataset (Figure 18, Appendix B.8).  With these experiments, we are confident that anticipatory recovery is a general phenomenon and not specific to the model or dataset we are using. We also want to point out that Pythia uses a very generic transformer architecture, and many experiments in the paper (Figure 2b, Figure 3, Figure 6a, Figure 14, Figure 15) uses randomly initialized Pythia models and is therefore not particular to the specific pre-training recipe of Pythia.
>
> **Re: “lacks citations for a substantial body of work on continual learning in LLMs”**
> > Thanks for pointing out these references. We will discuss these relevant works in the next version of the paper.

---

> > ### Comment · Reviewer_4gi1 · 2024-08-11
> > **Followup questions**
> >
> > Thank you for your detailed and thoughtful response. I understand the context you provided regarding cyclic training and its relevance to real-world scenarios where tasks repeat in predictable patterns. Your argument about the naturalistic sequence of tasks in day-to-day life is compelling and adds an interesting dimension to the discussion.
> >
> > However, drawing from my experience with deploying models in environments characterized by continual distribution shifts, I find that cyclic training, as described, may not be widely applicable in practical machine learning setups. While the phenomenon of anticipatory recovery is intriguing, I am still left questioning the broader implications of your findings.
> >
> > Specifically, what tangible contributions can a machine learning researcher or practitioner derive from this discovery? While the paper successfully highlights an interesting phenomenon in a particular training setup, it seems to stop short of demonstrating how this understanding can be applied in a meaningful way.
> >
> > I certainly appreciate the value of scientific discovery and advancing our understanding of model behavior, but I would argue that this understanding needs to translate into practical utility, especially when derived from a specialized setup like cyclic training. Can you provide more clarity on how anticipatory behavior could be beneficial in real-world applications? For example, could this phenomenon offer deeper insights into model generalization, particularly in scenarios with subtle cyclic patterns? How might your findings be leveraged when working with true agents in dynamic environments?

---

> > > ### Author Response · Authors · 2024-08-12
> > >
> > > We thank the reviewer for their response. We fully agree that cyclic training is still artificial and that it will be important to study the anticipatory recovery phenomenon in naturalistic task sequences that repeat only imperfectly or stochastically. We have made initial steps in that direction in this submission, including data randomization (Figure 4, lines 148-154), partial random shuffling (Figure 12, lines 602-608), and rebuttal experiment E3, which demonstrate that the anticipatory recovery phenomenon indeed generalizes to partially or stochastically repeating task sequences. The main focus of this paper, aside from identifying this surprising phenomenon, has been to understand its underlying mechanisms and what properties of models and learning algorithms it depends on. We hope you agree we have made good progress in this direction and that our findings will be valuable in guiding the next steps with naturalistic task sequences.
> > >
> > > The two questions at the end of your comment nicely point to two messages that were missing from our submission but that we will add to the revision.
> > > > Could this phenomenon offer deeper insights into model generalization, particularly in scenarios with subtle cyclic patterns?
> > >
> > > Our work suggests that the field of continual learning should pay more attention to naturalistic task sequences in which tasks interleave in statistically regular patterns.
> > > > How might your findings be leveraged when working with true agents in dynamic environments?
> > >
> > > The field might move beyond its focus on learning and retention of novel tasks to examine re-learning performance when old tasks recur. With the anticipatory recovery phenomenon, we identified an important way in which ML models can do surprisingly better than expected on this measure. By identifying the properties of models and learning algorithms that moderate this phenomenon, our experiments provide an important first step toward leveraging it with true agents in realistic dynamic environments.

---

> > > > ### Comment · Reviewer_4gi1 · 2024-08-13
> > > > **Increasing my score**
> > > >
> > > > I trust that you will address these limitations in your next revision and will increase my score accordingly.

---

### Official Review · Reviewer_dpPf · 2024-07-20

**Soundness:** 3
**Presentation:** 3
**Contribution:** 3
**Rating:** 8
**Confidence:** 4

**Summary:**

This paper takes a deeper look at the training dynamics of neural networks such as LLMs in a particular kind of non-IID learning setting when tasks or data is iterated through cyclically in a fixed repeated sequence. The authors find evidence in this setting of an emergent behavior that they call "anticipatory recovery" where the model improves its loss on data before it actually arises seemingly in anticipation of the upcoming data. This behavior is surprising because the neural network does not contain an explicit memory mechanism for this purpose. The authors run a number of experiments to provide further clarity about the nature of this phenomenon and present a simple toy setting to help explain how this may happen.

**Strengths:**

The main insight of this paper that neural networks of a certain size exhibit anticipatory recovery as an emergent property is a very novel and interesting concept to me. I had never considered that this kind of thing may be happening. As a continual learning researcher, this is definitely something that adds to my mental picture of what may be happening during continual training. There are really a very significant number of experiments in this paper and many of the first questions that would come to mind were answered by the authors. Throughout much of the beginning of the paper I was curious about what the potential causes of this behavior could be, so I really appreciate the analysis in section 4.4. It still is not very clear, but in my mind identifying the problem is a worthwhile contribution for the paper, so I was satisfied with the effort given this context.

**Weaknesses:**

The writing of this paper could be improved somewhat. Reading through the paper can feel like going along for a journey with the authors with very little idea of where it is headed ahead of time. Typically papers summarize the main findings upfront to a larger extent. A lot of the time it feels like a large set of results are presented with little explanation of their connections and then the authors add in a summary section to recap how everything ties together. It would be better to provide the summary up front so readers know where it is going.

The authors could also do a better job of contextualizing the potential implications of their work (see questions below). Moreover, the empirical results are somewhat limited by the choice to prioritize breadth in terms of the number of questions asked rather than depth in terms of coming to conclusive answers to these questions by running them over a number of datasets and architectures.

**Questions:**

Q1:  The authors seem to indicate that they always use a vanilla gradient descent optimizer. While it is obviously interesting to consider this case, I wonder whether this analysis is less relevant for typical methods of training LLMs i.e. with Adam optimization and cosine decay learning rate schedules. Do you have any thoughts on how this may impact recovery?

Q2: Do you believe that changing the total amount of data may change the nature of anticipatory recovery? I am also trying to understand the potential relevance to the one pass very large data setting. I guess due to shuffling, recovery is not expected despite redundancy?

Q3: I wonder if the authors see any actionable insights to follow up from this study. Are you actually advocating the cyclic training? Do you see any potential implications for continual learning or curriculum design?

**Limitations:**

The authors perform a number of experiments in the paper, which they use to draw various conclusions, but the authors could do a better job of noting the limitations in these experiments i.e. individual ablations only being run in single settings (which is understandable but still limits how much we could read into the results) and possible alternative explanations. Conceptually, the authors made the choice to consider a variety of experiments trying to answer a variety of questions rather than asking fewer questions and coming to more conclusive results about the answers to these questions.

---

> ### Author Rebuttal · Authors · 2024-08-07
>
> Thank you for the detailed feedback.
>
> **Re: Writing improvement and summary of main findings.**
> > We have a summary of main findings in the final paragraph of section 1 but we agree some elaboration will help readers anticipate what is coming. We thank the reviewer for the writing feedback, and will incorporate these improvements proposed by the reviewers in the next version of the paper.
>
> **Re: Q1: The analysis of Adam optimization and cosine decay learning rate schedules.**
> > We report an experiment comparing Adam to SGD which shows Adam produces a stronger anticipation effect in Figure 6 (lines 145-147) of the paper. We hypothesize this result fits a broader pattern in which the strength of the anticipatory recovery effect is related to how well the model can fit each successive training task (see Section 3.3). In rebuttal experiment E2 we experimented with the cosine learning rate schedule and verified that the model still exhibits strong anticipatory recovery.
>
> **Re: Q2: Total amount of data and the one-pass very large data setting.**
> > We report the effects of the total number of tasks in Figure 5a (lines 128-130) of the paper, and found that the anticipatory recovery effect is more significant with a higher number of documents in the sequence. The one-pass very large data setting should produce no anticipation, similar to what we find when the task order is fully reshuffled on each pass in figure 1a.
>
> **Re: Q3: Actionable insights to follow up from this study.**
> > Our paper challenges the conventional wisdom in the continual learning community on when catastrophic forgetting occurs: we suggest that for certain models, catastrophic forgetting can be largely mitigated when training on structured sequences. As argued in our general rebuttal, natural environments often have predictable transitions, and cyclic training is the most extreme version of this type of sequential structure. We therefore advocate for more research in structured training and rethinking classic continual learning algorithms in structured sequences, and using prequential evaluation as an important metric. We can also study the types of structures that achieve best learning performance and efficiency, which could contribute to designing good learning curricula and active learning methods.
>
> **Re: Limitations.**
> > We thank the reviewer for pointing out these limitations. We will mention these points in the limitations section in the next version of the paper.
> We also want to point out that, while many of the experiments are conducted on Pythia models and the CNN/Daily Mail dataset to identify the influential factors of the anticipatory recovery effect, we carried out many experiments with other models and datasets as well, including vision models (Figure 7, Section 3.4), the GPT-2-Large model (Figure 17, Appendix B.7), and wikitext-103 dataset (Figure 18, Appendix B.8). With these experiments, we are confident that anticipatory recovery is a general phenomenon and not specific to the model or dataset we are using. We also want to point out that Pythia uses a very generic transformer architecture, and many experiments in the paper (Figure 2b, Figure 3, Figure 6a, Figure 14, Figure 15) use randomly initialized Pythia models and are therefore not particular to the specific pre-training recipe of Pythia.

---

> ### Comment · Reviewer_dpPf · 2024-08-13
>
> I took some time to go through response to my review and the reviews by the other reviewers. I really appreciate the thorough response to my concerns and particularly value the new experiments analyzing the cosine decay learning rate schedule. I feel that this addition has improved the paper and made me even more confident about its relevance to the community. I thought a lot about it and decided to increase my score to an 8. Looking through the rating descriptions, the novel ideas aspect stood out to me.

---

### Author Rebuttal · Authors · 2024-08-07

**The practicality of cyclic training and relevance of prequential evaluation:**

​​We appreciate the reviewers' recognition of the technical soundness and overall quality of our manuscript. While reviewers dpPf and gu9E have appreciated the contribution of our findings, we understand that reviewers 4gi1 and 5Cpf raised concerns about the practicality of the cyclic training approach and the relevance of prequential evaluation. Here we provide a more comprehensive explanation of the practical applications and relevance of our findings. We will incorporate these into the next version of the paper.

We believe cyclic training and prequential evaluation are more natural and relevant for real-world agents than standard continual learning. In natural environments, tasks recur and they often do so in predictable patterns (e.g., a person anticipating to eat breakfast after taking a morning shower). Our cyclic training approximates this structure, while standard continual learning settings (e.g. task- and class-incremental learning) of seeing each task only once are artificial and fail to consider this structure. Furthermore, the only metric a real agent cares about is its performance on the tasks it actually encounters, at the times it encounters them. This is prequential evaluation, which examines the quality of predictions at each time step conditioned on knowledge from earlier experience.

The aim of our paper is not to propose cyclic training as a method to improve performance on standard continual learning metrics. Instead we argue many natural environments are quasi-cyclic and so it is important to understand how existing deep learning methods perform in this setting. Anticipatory recovery is a surprising and significant discovery in this regard.

**New Experiment Results on pre-training, cosine learning rate schedule, and structured training:**

**Experiment E1:** In response to question 6 from reviewer gu9E, we took Pythia models pre-trained for a different number of steps (6K, 12K, 24K, 48K, 96K), and we found that more pre-training does give rise to higher anticipatory recovery.  As we summarize at the end of section 3.3, we hypothesize this result fits a broader pattern in which the strength of the anticipatory recovery effect is related to how well the model can fit each successive training task. Models with more pre-training steps are more capable of fitting each successive training task, and therefore exhibit higher anticipatory recovery. The experiment results can be found in Figure 1 of the rebuttal PDF.

**Experiment E2:** In response to question Q1 from reviewer dpPf, we experimented with cosine learning rate scheduling on the Pythia-1B model with 10 epochs. We show that the model also exhibits the anticipatory recovery effect in typical LLM optimization schemes. The experiment results can be found in Figure 2 of the rebuttal PDF.

**Experiment E3:** In response to a question from reviewer 5Cpf, we experimented with a new setting where only the first 20 documents are kept fixed in each epoch, and a random number of other documents are inserted after the first 20. This new setting generalizes cyclic training in that (1) rather than having the same documents in every epoch, we insert random other documents between every repetition (2) epochs can have different lengths. We still observe anticipatory recovery for documents 2 through 20 in this setting, suggesting that anticipatory recovery exists as long as there is a repeating sub-sequence in the data stream. The experiment results can be found in Figure 3 of the rebuttal PDF.

---

### Decision · Program_Chairs · 2024-09-25

**Decision:**

Accept (poster)

**Comment:**

This paper studies a specific kind of structured training dubbed cyclic training, where documents are presented cyclically in a fixed, repeated sequence. This reveals that LLMs exhibit anticipatory recovery behavior, i.e. they begin to recover from forgetting a document before encountering it again in the sequence. Some insights into the training dynamics in the cyclic training setup are offered, emphasizing that this anticipatory behavior is not exclusive to LLMs but can be observed in other large-scale networks as well. Reasons for this anticipatory recovery are investigated: finding initial evidence that suggests adjacent documents in the training sequence are represented more similarly by the model.

Scores include strong accept (8), accept (7),  borderline accept (5), and weak reject (4). Despite this range of scores, the reviewers fundamentally agree on:
- Strength: what makes this problem exciting is  that  it “identifies a surprising phenomenon where models exhibit anticipatory recovery, which is counter-intuitive given the understanding of how LLMs handle sequential knowledge without explicit memory mechanisms” (reviewer dpPf who gave a 8)
- Weaknesses: there are serious “significance” issues (some rate it in the contribution score). As stated by reviewer 5Cpf who gave a 4: “The main weakness of the paper is its unclear connection with common practices in training neural networks, therefore, it can be a little bit difficult to appreciate the significance of the findings”

To dig out further in the issue of the significance of this work:
- Could it lead to improved "learning curricula that achieve learning efficiency “?
 However, several reviewers do not see much future in the fixed order training required by the method “training with fixed example order usually achieves worse generalization performance than random-shuffling”.
- Could it help our theoretical understanding of the training dynamics of neural networks? However, this paper only brings a little light to this understanding.

In summary, the reviewer disagreement is about how critical the paper should be to the ML practitioner, which represents nowadays a large fraction NeuRIPS attendance. As a long time NeuRIPS attendee, I don't think it should be limited to the practitioner, and even a paper whose main contribution is to challenge our understanding of the learning process is worth publishing, so I am inclined towards accept.